# Community Perceptions of HIV Stigma, Discriminatory Attitudes, and Disclosure Concerns: A Health Facility-Based Study in Selected Health Districts of South Africa

**DOI:** 10.3390/ijerph20146389

**Published:** 2023-07-18

**Authors:** Mathildah Mokgatle, Sphiwe Madiba

**Affiliations:** 1Department of Public Health, School of Health Care Sciences, Sefako Makgatho Health Sciences University, Pretoria 0204, South Africa; mathildah.mokgatle@smu.ac.za; 2School of Transdisciplinary Research and Graduate Studies, College of Graduate Studies, University of South Africa (UNISA), Muckleneuk, Pretoria 0001, South Africa; 3Faculty of Health Sciences, Executive Deans Office, University of Limpopo, Polokwane 0700, South Africa

**Keywords:** disclosure concerns, discrimination, HIV related stigma, South Africa, interventions, people living with HIV (PLHIV)

## Abstract

Research data about HIV stigma perceptions and discriminatory attitudes among the general population are limited. Furthermore, the willingness of HIV-negative individuals to engage with HIV prevention and disclosure interventions has not been established in South Africa. The study investigated community perceptions of stigma as well as discriminatory attitudes towards HIV disclosure to understand if and how these perceptions might influence the uptake of disclosure interventions. This facility-based study used a validated questionnaire to measure the four constructs of HIV stigma among 670 adults recruited from health districts of two provinces of South Africa. Of these, 72% were female, 87% had ever been tested for HIV, and 31% knew someone who has HIV. Stigma towards people living with HIV (PLHIV) is widespread in the general population. A high proportion (75%) endorsed disclosure concerns, 75% perceived stigma to be common, and 56% endorsed negative statements indicating perceived stigma in communities. Fear, moral and social judgement, and rejection underlined their perceptions about PLHIV. Almost half (45.7%) were unwilling to care for family members sick with AIDS, suggesting negative distancing reactions and discriminatory attitudes towards PLHIV. The widespread discriminatory attitudes and the perceived stigma that is evident in the general population might heighten the disclosure concerns endorsed, promote non-disclosure, and increase HIV transmission. To design interventions, it is crucial to be cognisant of disclosure concerns, discriminatory attitudes, and perceived stigma evident in communities. Thus, the findings underscore the need to increase efforts to challenge and reduce community drivers of negative discriminatory attitudes and perceived stigma.

## 1. Introduction

The Joint United Nations Programme on HIV/AIDs (UNAIDS) asserts that although HIV-related stigma has been declining in Sub-Saharan Africa (SSA) it remains high in several countries [1]. People living with HIV (PLHIV) residing in low- and middle-income countries (LMICs) are at greatest risk of HIV-related stigma [2,3]. South Africa has the world’s worst epidemic [4] with 7.8 million South Africans estimated to be living with HIV (LHIV) [4]. Thus, PLHIV residing in the South Africa are at high risk of experiencing HIV-related stigma [5,6]. A systematic review of South African studies found rates of internalised stigma ranging from 22% to 41%, the rate of anticipated stigma ranging from 24.4% to 43%, and the prevalence of any stigmatising experience ranging from 43.5% to 88% [6].

Within the context of HIV, stigma may be best understood as a combination of “enacted stigma” (the discrimination, discounting, discrediting, devaluation, stereotyping, and/or prejudice by others because of one’s HIV status), anticipated stigma (being aware of how society views HIV, and an expectation of how one will be treated if one’s status is disclosed or becomes public), internalised or self-stigma (the adoption of a negative self-image and devaluation that results from living with HIV), and perceived stigma, also known as community stigma (perceptions of the existence and severity of stigmatising attitudes in the community that lead to anticipated stigma) [7,8,9]. Evidence suggests that perceived stigma, or the fear of being stigmatised, is the most prevalent form of stigma in the era of antiretroviral expansion [10].

HIV stigma is aggravated by variation of the effects of stigma across cultural and social groups [6,8,11], hence the pockets of HIV-related stigma observed in many geographic settings within SSA. Thus, stigma is experienced in different ways and at different rates even within geographic locations [8,10,12]. For instance, some subgroups experience more severe forms of stigma and discrimination than others. Globally, research studies suggest that adolescents with perinatal HIV (APHIV), HIV-positive pregnant women, and older people are more likely to report all forms of stigma [3,5,13,14,15]. However, research studies across SSA have shown that adolescents with perinatal HIV (APHIV) are at greater risk of HIV-related stigma than adults LHIV [5,16,17]. In addition, APHIV are likely to report most forms of stigma [5,15,18] due to associative stigma arising from their mothers’ HIV status.

The consequence of persistent high levels of anticipated stigma within communities is PLHIV hiding their HIV status from others [7,8]. All forms of stigma have been repeatedly linked with decreased voluntary HIV testing, non-disclosure of HIV status, and unwillingness to disclose [19,20,21,22]. Several research studies identified internalised stigma as a key risk factor for more severe negative outcomes than other forms of stigma amongst APHIV, including non-adherence to antiretroviral therapy (ART), disengagement with care, non-disclosure, and morbidity [13,23,24]. Internalised stigma was also found to be a stronger predictor of depressive symptoms than other forms [6,14,23,25]. Meanwhile, enacted stigma has been associated with poor physical health and increased stress response. Research studies show that enacted stigma can lead to development of internalised stigma, thus negatively impacting quality of life of PLHIV. Enacted stigma is also one of the key barriers to access and utilisation of health care and adherence to medication in many settings [9,26,27].

Research shows that increased access to ART has not changed environmental factors contributing to low rates of disclosure and unwillingness to disclose among PLHIV. In Uganda, a significantly higher percentage of people in the general population endorsed fears about disclosure due to an increase in anticipated stigma [28]. It has been argued that disclosure occurs when PLHIV perceive the environment to be safe to disclose [29]. To avoid being identified as HIV-positive and reduce the likelihood of being stigmatised, the majority of PLHIV are unwilling to disclose to others [10,21,30,31]. In the era of treatment expansion, fear of unintended disclosure leads to PLHIV disengaging from care or skipping doses if confidentiality is at risk. As such, unintentional disclosure through medication collection and use has been identified as a new HIV-related stigma that impedes adherence and retention in care services for both adults and adolescents LHIV [10,16,32,33]. 

While the benefits of disclosure have been reported since the beginning of the HIV and AIDS epidemic, the desire to maintain silence about an HIV test persists in many societies including among APHIV [16,17,34,35,36]. The normalisation of non-disclosure or secrecy about HIV diagnosis by people affected by and infected with HIV makes disclosure even more complex and consistently drives HIV-related stigma [23]. Though self-disclosure leads to early initiation of ART, improved treatment adherence, and access to psychosocial support [35,37,38], on the other hand, it also leads to stigma. For instance, self-disclosure raises several concerns for APHIV including the possibility of rejection, social isolation, losing friends, social exclusion, diminished social interactions, and ostracism [22,35,39]. The perceived enactment of stigma has been linked to the reason for non-disclosure among APHIV [15,37,39,40].

Despite the low disclosure rates and persisting stigma in communities across SSA, community-related factors that contribute to low disclosure levels are not well researched in many settings in the region, including in South Africa. Data concerning experiences and perceptions of HIV-related stigma have been obtained in specific populations [41]. On the contrary, research data about the HIV stigma perceptions, attitudes, and discrimination among the general population in South Africa and other SSA countries with high HIV prevalence are limited [42]. Furthermore, research on the acceptability of disclosure in the general HIV-negative population is limited and not well defined and yet it is critical. This explains the dearth of data on the role of the community in enabling or inhibiting disclosure. These kinds of data are necessary to inform disclosure intervention designs that are context and culturally specific for children and APHIV to target and reduce internalised stigma [6,23].

The aim of this study was to examine community perceptions of stigma as well as discriminatory attitudes towards PLHIV and measure disclosure concerns to understand if and how these perceptions might influence the uptake of HIV disclosure interventions. The willingness of the general population of HIV-negative individuals to engage with HIV prevention and disclosure interventions has not been established in South Africa. Furthermore, systematic reviews that examine HIV stigma reduction interventions to reduce HIV stigma particularly among APHIV in LMICs are limited [11]. Stangl and colleagues reviewed 48 stigma reduction interventions. While only three studies were aimed at PLHIV in SSA, none of these interventions focused on stigma reduction in APHIV [43]. Furthermore, Tran and colleagues reported a lack of empirical studies on the design of interventions for specific contextual factors [44]. In the context of HIV prevention and disclosure interventions, it is crucial to measure HIV stigma among PLHIV and community members in areas where the interventions are being implemented [45].

Therefore, the secondary objective of the study was to examine community acceptability of HIV disclosure interventions within the context of piloting a disclosure model and intervention materials in selected health facilities in two provinces of South Africa. We used recommendations from numerous research studies to develop and implement community-based interventions to correct community misperceptions about HIV and disclosure [6,22,23,46]. It is crucial that interventions aim at improving disclosure and reducing internalised sigma in children and APHIV to avert the negative psychological outcomes associated with stigma and non-disclosure [20,46].

## 2. Materials and Methods

### 2.1. Study Design and Setting

This cross-sectional quantitative survey was conducted with adult clients visiting health care facilities for scheduled appointments. While the investigators intended to conduct a community-based study, the absence of suitable community-based infrastructure to do so safely and effectively led to using health facilities instead. Thus, respondents were recruited from three health facilities located in Tshwane health district in Gauteng and rural Nkangala district in the province of Mpumalanga in South Africa. These facilities had previously been included in studies conducted by the investigators to examine disclosure to children and adolescents with HIV from the perspective of various study populations [35,47]. The health facilities were sites to pilot a disclosure intervention for children and adolescents with HIV. The data presented here are part of a larger study to develop, implement, and assess a counselling-based disclosure model to guide health care workers to prepare and support care givers of children with PHIV in the disclosure process (intervention not reported here). This paper presents the levels and contexts of HIV stigma and disclosure concerns among communities (adult clients) where the intervention would be piloted.

### 2.2. Study Population

Adults 18 years and above, independent of their HIV status, were eligible to participate in the survey that we conducted between February and April 2018. A sample of 670 adult clients was obtained using Cochran’s formula assuming a confidence level of 95% and margin of error of 0.05. Respondents were sampled through systematic random sampling since all clients in the waiting areas of the selected health facilities were eligible to participate.

### 2.3. Materials, Methods, and Measures

The research team comprising trained fieldworkers introduced the study to eligible adult clients who visited the facilities during the study period. Participation was solely voluntary and those who consented to take part responded to the questionnaire at the facilities. Completion of the questionnaire was carried out in a private room assigned to the research team to allow respondents to answer sensitive questions privately. This was crucial since the questions asked were about personal and sensitive issues regarding stigma and discriminatory attitudes towards PLHIV.

An anonymous self- or researcher-administered questionnaire was administered depending on the literacy levels of study respondents. The questionnaire captured a range of variables including demographic information, history of prior HIV testing, knowing someone with HIV, disclosure concerns, acceptability of disclosure interventions, and HIV-related stigma. The questionnaire was translated into IsiZulu and Setswana since these were the spoken languages in the setting.

The primary outcomes were: (1) discriminatory attitude towards PLHIV, (2) disclosure concerns, and (3) HIV-related stigma (internalised and perceived stigma). To measure the 4 constructs of HIV-related stigma we used a 27-item scale derived from several different sources that had been shown to have good internal consistency, reliability, and construct validity in a number of studies conducted with PLHIV [7,42,48,49]. The 27-item scale measured perceived stigma, public attitudes/discriminatory attitudes, personalised stigma/negative self-image, and disclosure concerns. Respondents were asked to indicate the extent to which they agreed with statements about PLHIV. To simplify analysis, we modified responses to agree, disagree, or not sure instead of a four-point agreement rating scale from strongly agree to strongly disagree. We coded the answers dichotomously as 1 = agree, 0 = disagree or not sure [7]. The total scale scores were created by summing responses across the items and higher scores indicate a greater degree of stigma and stigma constructs. 

#### 2.3.1. Perceived Stigma

Seven items were used to assess perceived community stigma. Participants were asked questions about their perception of other people’s reaction to PLHIV, reflecting the perceived community stigma. Sample items included “PLHIV are treated differently”, “people think less of someone because they have HIV”, “most people are uncomfortable around someone with HIV”, “people’s attitudes about HIV make PLHIV feel worse about themselves”, “people avoid touching people who are HIV positive”, “people who are HIV positive lose friends after sharing status”, and “people who care about HIV positive people stop phoning them after disclosure“. Respondents were considered to have perceived stigma if they thought that people treat PLHIV badly. Higher scores on this scale indicate a greater degree of perceived community stigma. Perceived stigma scores ranged from 0–7.

#### 2.3.2. Internalised Stigma

Four items selected from Kalichman’s measure of community-held AIDS-related stigma in the South African general population [6] captured internalised stigma. We rephrased the wording to reflect community perceptions of negative self-image and devaluation that result from living with HIV. The items included “people who are HIV positive feel guilty”, “people with HIV are ashamed of themselves”, “people with HIV don’t feel as good as other people because they are positive”, and “people’s attitudes about HIV make people with HIV feel worse about themselves”. Scores range from 0–4, with higher scores indicating greater perceptions of internalised stigma among PLHIV.

#### 2.3.3. Disclosure Concerns Scale

Disclosure concerns include worrying about negative outcomes of disclosure and trying to keep HIV status a secret [45]. Five items were used to assess HIV disclosure concerns. Disclosure items included “telling someone you are HIV positive is risky”, “people who are HIV positive work hard to keep their status a secret”, “people who are HIV positive are careful who they tell their status to”, “if a family member tests HIV positive, would you want it to remain a secret?” and “people with HIV should disclose their HIV status to others”. Disclosure concern scores range from 0–5. Higher scores indicated greater disclosure concerns.

#### 2.3.4. Discriminatory Attitudes towards PLHIV

Eleven items adapted from previously developed scales to measure HIV-related stigma [42,49,50] were used to assess perceptions of discriminatory actions and attitudes faced by people with HIV in communities. Sample items included “people with HIV have themselves to blame”, “I am afraid to be around people with HIV”, “most people are uncomfortable around someone with HIV”, “unwilling to drink from a tap if a person with HIV has just drunk from it”, “unwilling to have a relationship with a person if I know that he/she has HIV”, “unwilling to buy food from a food seller who has the AIDS virus”, and “unwilling to care for family members who become sick with AIDS”. Attitude scores range from 0–11. Higher scores indicated a more discriminatory attitude.

### 2.4. Data Analysis

Statistical analyses were performed using the STATA statistical software package version 17.0 (Stata Corp, College Station, TX, USA). Pearson chi-squared tests were performed to generate frequencies, proportions, and distributions of the sociodemographic variables. To measure attitudes towards PLHIV, internalised stigma, disclosure concerns, and perceived stigma, we coded the answers dichotomously by assigning a score of one (1) for agreeing with a negative statement or not sure and zero (0) for disagreeing with a negative statement. We then calculated the mean scores, and scores equal to or more than the mean were categorised as highly stigmatising, discriminatory, or negative attitudes. The paired *t*-test was used to compute the mean stigma scores and they are presented as the mean (standard deviation).

### 2.5. Ethical Approval

Ethical approval for the study was obtained from the Research Ethics Committee of Sefako Makgatho Health Sciences University (SMUREC/H/109/2016: IR). Prior permission to conduct this study was obtained from the relevant provincial, district, and facility authorities. Respondents provided written informed consent, participation was voluntary, and confidentiality of responses was maintained by not collecting any personal identifiers. At analysis, a unique identifier was assigned to every questionnaire.

## 3. Results

### 3.1. Demographic Charecteristics of Study Sample

The majority of the sampled 670 respondents were female (72%), the mean age was 27 years, and the range was 28 to 58 years. The majority (83.2%) were single, 68.8% were unemployed, and 87% had ever been tested for HIV. Only about a third (31%) knew someone who has HIV (Table 1).

### 3.2. Perceived Stigma

Perceived community stigma reflects the general perception of HIV and AIDS in a community [47]. Perceived community stigma and discrimination captured the extent to which respondents believe that people with HIV are unfairly treated or are subjectively aware of the extent to which others in the population harbor negative attitudes toward PLHIV [46]. Three quarters (75.6%) of the respondents endorsed the statement that PLHIV experience stigma and are treated differently in their communities (55%). Almost half (45%) were in agreement that most people are uncomfortable around PLHIV and 43% avoid touching them. Forty percent agreed that PLHIV lose friends after sharing status (Table 2).

### 3.3. Community Discriminatory Attitudes toward PLHIV

Although most respondents (92.2%) indicated that PLHIV deserve the same respect as everyone else, fear and moral judgement underlined their perceptions. Concerning moral judgement towards PLHIV, small proportions responded with negative responses associated with blame or shame towards PLHIV. About 11.6% endorsed the statement that PLHIV have themselves to blame and should be ashamed of themselves (4.2%). A significant proportion responded with negative responses associated with social distancing towards PLHIV. Almost half (45.9%) would not have a relationship with an HIV-positive person and were unwilling to care for family members sick from HIV (45.8%) (Table 3).

### 3.4. Internalised Stigma

Table 4 presents statements reflecting community perceptions of negative self-image and devaluation that results from living with HIV as well as disclosure concerns. Slightly less than half (44%) endorsed the devaluation statement that PLHIV feel worse about themselves and over a third (37%) agreed that PLHIV feel guilty about their HIV status. A significantly high percentage of respondents endorsed fears about disclosure. About three quarters (73%) endorsed the belief that disclosing HIV serostatus is risky, 86.6% agreed that PLHIV are very careful as to whom they tell their status, and 41% would want HIV infection in family to remain secret.

### 3.5. Disclosure Acceptability

Disclosure acceptability was relatively low, as only 60.2% of the respondents agreed that disclosure is acceptable and 69.7% agreed that PLHIV should disclose. A high proportion (81%) endorsed disclosing to sexual partners, disclosing to children with perinatal HIV (84.5%), and disclosing parents’ own HIV status to children (82.3%). Disclosure to non-family members was not acceptable (48.8%). Almost all (94%) respondents would encourage participation in disclosure interventions (Table 5).

### 3.6. Overall Stigma Scale

Respondents indicated a significant degree of stigmatisation in the community, with a moderately high mean perceived stigma score of 2.9 (SD = 1.8) and over half (56.8%) of the respondents endorsing moderately high levels of perceived stigma in communities. On the other hand, disclosure concerns were very high with 75% of respondents endorsing disclosure concerns. The mean disclosure concern score was 3.1 (SD = 1.2). For community discriminatory attitudes, the stigma score was 1.5 (SD = 1.6) and for internalised stigma the score was 1.5 (SD = 1.6), considered as representing low stigmatising attitudes (Table 6).

Overall, respondents perceived stigma in the community to be high. The mean overall stigma score across all constructs was 9.1 (SD = 4.4). Three quarters (75.6%) of the respondents perceived stigma towards PLHIV to be high in communities.

## 4. Discussion

We investigated levels and contexts of HIV stigma and disclosure concerns and established that stigma towards PLHIV is widespread in communities both in rural and urban settings. This is in spite of more than two decades of intensive ART provision and the launch of a national stigma reduction campaign by the South African government [7]. We found that a high percentage (75%) of respondents endorsed disclosure concerns followed by perceived stigma (55%). While overall the stigma scale scores were moderately high, 75% of respondents perceived stigma towards PLHIV in the community to be high and common. Similarly, Visser compared HIV-related stigma in South Africa between 2004 and 2016 and concluded that perceived community stigma has remained high [51], while findings of a rapid situation analysis conducted with multiple data sources in South Africa also found that perceived stigma persists [10]. 

The current findings are consistent with qualitative data from a South African rural community which established persistent levels of HIV-related stigma [52]. Prior studies also established that, while enacted stigma had reduced considerably, perceived community stigma has remained high [28,51,52,53]. Similarly, data from rural Uganda [50], Ethiopia [54,55], Nigeria [56,57], and elsewhere in Uganda [28,58] and qualitative data from Kenya and Uganda [53] reported the persistence of HIV-related stigma in the general HIV-negative population. These settings, like South Africa, had experienced the HIV epidemic for over two decades, however, it remains unclear why perceived stigma has remained widespread in the era of ART. Respondents in the current study, as in other studies [10,53], reported that gossip, mockery, distancing, and fear of infection persisted, despite intensive ART provision which has led to restored physical health of growing numbers of PLHIV [28]. Majola and colleagues argue that concealability of HIV does not lead to a decline in stigma but that PLHIV can avoid stigma if they cannot be identified [52]. Lack of comprehensive knowledge about HIV transmission and fear of HIV infection might explain the persistent perceived stigma in communities.

We further found evidence of stigma and discriminatory attitudes in the current study albeit the mean score is relatively low (1.5 (SD = 1.6)) and considered as representing low stigmatising attitudes. A recent study assessing discriminatory attitudes towards PLHIV among the general population in 15 nations in SSA found high prevalence of discriminatory attitudes towards PLHIV [59]. High levels of discriminatory attitudes towards PLHIV were also found in Uganda [58], Democratic Republic of Congo and Nigeria [57], Saudi Arabia [60], and Thailand [61]. In the current study, almost half (44.7%) of the respondents were unwilling to care for a family member sick with HIV and AIDS. This is in contrast with studies from Nigeria [56,62] and Uganda [58] where up to 70% of people were willing to care for a household member with HIV. Unwillingness and reluctance to care for and treat family members sick with HIV and AIDS are evident in the current study and in others, suggesting high levels of distancing reactions, stigma, and discriminatory attitudes towards respondents’ own family members with HIV. This explains the reluctance of PLHIV to disclose HIV serostatus to family members [30,56,63,64,65,66]. The fear of caring for a family member sick with AIDS further indicates the fear of HIV infection that results from lack of knowledge.

Regarding social distancing towards PLHIV, in this study a third (30.7%) of the respondents were unwilling to have relationships with PLHIV. Yet, significantly lower proportions were unwilling to buy from a food seller who has HIV, drink from the same tap as a person with HIV, and be friends with someone with HIV. Visser suggests that the general population regard close personal contact such as dating a person with HIV and having to care for family with HIV and AIDS as threatening [51]. In the South African study by Visser, the number of participants who were unwilling to buy vegetables from a vendor with HIV was significantly lower compared to those who were unwilling to care for a family member with HIV and AIDS. In a trend analysis in Nigeria, however, those who were willing to buy food from sellers with HIV and AIDS was fewer compared to those who were willing to care for family with HIV and AIDS [58]. The willingness to care for family with HIV and AIDS implies some level of empathy for family members with HIV [51], whereas they expressed aversion towards food vendors with HIV [54,55,61,62], signifying serious levels of distancing and discrimination towards PLHIV. It is therefore important to implement strategies and programmes to increase HIV knowledge and promote compassion and respect for the rights of PLHIV so as to eradicate discriminatory attitudes and practices towards PLHIV [56].

We found that fear and moral and social judgement also underlined the perceptions of respondents about PLHIV. At the community level, stigma may include blame or shame which constitutes negative judgments of PLHIV and distancing practices for PLHIV [42]. In the current study, a relatively small proportion of respondents endorsed the belief that PLHIV should be ashamed and blamed for contracting the disease. On the other hand, researchers in Nigeria [56], Ethiopia [54,55], and Thailand [61] found that community members not only felt that it was shameful to be infected with HIV, but that PLHIV should be blamed for bringing the disease to the community. This is not the case in South Africa, where women in particular are often blamed by their sexual partners for bringing the infection to the household. Thus, PLHIV often experience avoidance behaviours from members of the communities where they reside. For instance, a study conducted in Saudi Arabia found that over half of the general population were unwilling to live with a friend who has HIV and AIDS [60]. Therefore, evidence from the current study and previous studies points to moderate to high levels of rejection of PLHIV by the general population on the basis of blame and prejudice.

We further found significant concerns for disclosure in the general population. While almost all the respondents had positive attitudes towards participation in disclosure interventions, a significantly high percentage (73%) endorsed the belief that disclosing ones’ HIV status is risky. The findings are comparable with disclosure concerns reported among general populations in prior studies [28,67] and PLHIV [68,69]. Furthermore, slightly less than half (41%) of the respondents were unwilling to reveal a family member’s positive HIV status and would encourage family members LHIV to remain silent about their HIV status. Prior studies reported similar findings [49,53,55]. Disclosure is perceived as risky due to the high levels of perceived stigma among communities. The disclosure concerns raised in the study are an indication that discrimination attitudes against PLHIV are persistent, similar to what was reported elsewhere [54,58,59] and give credence to prior studies that reported low disclosure rates in various settings in SSA. Non-disclosure is often an action taken to prevent stigmatisation, being labelled, and being defined by HIV [29,63,70,71]. 

There is substantial evidence that stigma and discriminatory attitudes towards PLHIV reinforce concealment of HIV status, leading to secrecy, taking steps to avoid being identified as HIV positive, and denial [10,21,30,31]. Disclosure concerns and a low rate of disclosure have been identified as major concerns in tackling the HIV epidemic [58]. Thus, the Ending AIDS Strategy [72] would not work if people worry about the stigma they would face if they were HIV positive [61]. Kenworthy and colleagues argue that ending AIDS must be premised on ending fear, stigma, discrimination, and other relevant conditions that fuel infections [73]. Furthermore, HIV-related stigma and discrimination directly impede adherence to ART [9,32,74], as medication collection has been linked with unintentional disclosure, a new HIV-related stigma [32,69].

One of the limitations of this study is that we utilised health facilities to conduct a community study due to the absence of suitable community-based infrastructure. Therefore, it is probable that people who do not use health services are more likely to hold different negative views toward PLHIV and disclosure concerns or are even less likely to be aware of their HIV status. Secondly, our study relied on self-reported measures, which are susceptible to reporting biases, particularly for discriminatory attitudes and stigma towards PLHIV. Thirdly, we cannot generalise the findings to the general population as the study was facility based. Nevertheless, the findings confirm the suggestion of persistent stigma towards PLHIV in the era of expansion of ART. Lastly, the study used a cross-sectional design, and therefore we cannot establish stigma and discriminatory attitudes over time.

## 5. Conclusions

We found evidence that stigma and discriminatory attitudes are persistent in rural and urban communities in South Africa. Community members also view stigma and discriminatory attitudes toward PLHIV as common occurrences. The moderately high levels of stigma that we identified might heighten the perceived threat of community discriminatory and stigmatising attitudes towards PLHIV, promote non-disclosure of HIV status, and increase HIV transmission. The findings further pointed out the need to strengthen stigma reduction efforts in the national HIV response.

The disclosure concerns raised are consistent with the discrimination attitudes against PLHIV and perceived stigma that are evident among community members. While disclosure interventions are accepted by the general population, to design and implement disclosure interventions it is crucial to be cognisant of discriminatory attitudes towards PLHIV and perceived stigma evident among community members. Of note is that the purpose of disclosure interventions is not only to increase disclosure across population groups but also to prepare the environment for disclosure to occur early after infection without fear of stigma and discrimination. Thus, the study findings underscore the need to increase efforts focused on challenging and reducing community drivers of negative discriminatory attitudes that currently rank low in the priority list of HIV programmes. Reducing stigma is crucial for South Africa to attain the goals of the Ending AIDS Strategy to prevent new infections to end AIDS given the role stigma plays in HIV transmission.

## Figures and Tables

**Table 1 ijerph-20-06389-t001:** Sociodemographic characteristics of respondents.

Variables	Frequency (%)
Gender	
Male	182 (28.0)
Female	467 (72.0)
Age	
18–25 years	262 (39.7)
26–50 years	384 (58.2)
51+ years	12 (2.1)
Marital status	
Ever married	111 (16.8)
Single	550 (83.2)
Employment	
Employed	207 (31.2)
Unemployed	456 (68.8)
Income source	
No income	256 (43.5)
Grant	153 (26.0)
Partner/family support	27 (4.6)
Employed	153 (26.0)
Ever tested for HIV	
Yes	580 (87.5)
No	83 (12.5)
Knowing someone with HIV	
Yes	196 (31.2)
No	366 (58.4)
Not sure	65 (10.3)

**Table 2 ijerph-20-06389-t002:** Proportion of respondents agreeing with perceived stigma beliefs.

Perceived Stigma Statements	Agree n (%)
HIV-positive people experience stigma from the community	498 (75.6)
People with HIV are treated differently	365 (55.1)
Most people are uncomfortable around someone with HIV	312 (46.9)
People avoid touching people who are HIV positive	280 (43.0)
People who are HIV positive lose friends after sharing status	271 (40.8)
People who care about PLHIV stop phoning them after disclosure	109 (16.9)
People think less of someone because they have HIV	48 (7.2)

**Table 3 ijerph-20-06389-t003:** Proportion of respondents agreeing with discriminatory attitudes.

Discriminatory Attitudes Statements	Agree n (%)
I will not have a relationship with an HIV-positive person	305 (45.9)
Unwilling to care for HIV-positive family member	304 (45.7)
People with HIV have themselves to blame	77 (11.6)
I feel uncomfortable around people with HIV	50 (7.5)
Getting HIV is a punishment for bad behaviour	33 (5.0)
Unwilling to buy food from an HIV-positive food seller	34 (5.1)
Unwilling to drink from a tap if a person with HIV has just drunk from it	32 (4.8)
People with HIV must have done something wrong to deserve it	30 (4.5)
People with HIV should be ashamed of themselves	28 (4.2)
I am afraid to be around people with HIV	25 (3.8)
Unwilling to be friends with someone with HIV	22 (3.3)

**Table 4 ijerph-20-06389-t004:** Proportion of respondents agreeing with internalised stigma statements and disclosure concerns beliefs.

Internalised Stigma Statement	Agree n (%)
People who are HIV positive feel guilty	246 (37.2)
People’s attitudes about HIV make PLHIV feel worse about themselves	292 (44.1)
PLHIV do not feel as good as other people	246 (37.8)
People with HIV are ashamed of themselves	18 (2.7)
Disclosure concerns beliefs	
People with HIV are very careful to whom they tell their status	575 (86.6)
Telling someone you are HIV positive is risky	487 (73.1)
People with HIV should disclose their status to others	458 (69.7)
People who are HIV positive work hard to keep their status a secret	460 (69.1)
If a family member is positive, I would want it to remain a secret	270 (41.0)

**Table 5 ijerph-20-06389-t005:** Proportion of respondents who agreed with disclosure statements.

Acceptability of Disclosure and Intervention Statements	Yes/Agree n (%)
Would you encourage family member to participate in disclosure interventions?	616 (94.0)
Is it ok for parents/caregivers to tell the child about his/her HIV-positive status?	557 (84.5)
Is it ok for parents to disclose their own parental HIV status to children?	542 (82.3)
Is it ok for HIV-positive people to disclose to sexual partners?	533 (81.0)
Should people who are HIV+ disclose their HIV status?	458 (69.7)
Is it ok for parents/caregivers to disclose status of a child to other family members?	414 (62.8)
Is HIV disclosure acceptable among HIV-negative people in the community?	374 (60.2)
Is it ok for parents/caregivers to disclose child status to non-family members?	338 (50.9)
Is it ok for HIV-positive people to disclose to non-family members?	322 (48.8)

**Table 6 ijerph-20-06389-t006:** Descriptive statistics for the stigma scales.

Stigma Constructs	Score Range	Mean (Standard Deviation)
Perceived stigma	0–7	2.9 (1.8)
Disclosure concern	0–5	3.0 (1.2)
Community discriminatory attitudes	0–11	1.5 (1.6)
Internalised stigma	0–4	1.5 (1.6)
Overall stigma	0–27	9.1 (4.4)

## Data Availability

The data that support the findings of this study are available from the corresponding author, upon reasonable request.

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
