# Peer review of "Community Perceptions of HIV Stigma, Discriminatory Attitudes, and Disclosure Concerns: A Health Facility-Based Study in Selected Health Districts of South Africa"

_ijerph, 2023, doi:10.3390/ijerph20146389_

Round 1

Reviewer 1 Report

1. The title is long. Please

2. < 250 words

3.< 250 words in abstract

4. PLHIV in keywords

5. PLHIV lin 81

6. There are many tables

7. Please refer to

Antiretroviral therapy adherence and its determinant factors among people living with HIV/AIDS: a case study in Iran  in discussion.

8. Use more up-to-date references

Community Perceptions of HIV Stigma is important in all communites and all people must be learning and act.

It is good

Author Response

We thank the reviewer for taking time to review and give valuable comments 

Reviewer 2 Report

Comments for authors:

This paper reports important findings on HIV stigma and discrimination. My comments for improvement are listed below:

Abstract:

Research data about HIV stigma perceptions and discriminatory attitudes among the general population are limited.

·        Please revise this sentence, there have been a massive number of studies about HIV stigma and discrimination around the world.

Introduction:

“People living with HIV (PLHIV) residing in low and middle-income 39 countries (LMICs) are at greatest risk of HIV-related stigma [2].”

·        Please consult and use the following current systematic literature review to support this claim:

Psychological and Social Impact of HIV on Women Living with HIV and Their Families in Low- and Middle-Income Asian Countries: A Systematic Search and Critical Review. International Journal of Environmental Research and Public Health. 2022; 19(11):6668. https://doi.org/10.3390/ijerph19116668

“Within the context of HIV, stigma may be best understood as a combination of “enacted stigma” (negative beliefs one has about oneself), anticipated stigma (being aware of 47 how society views HIV, and an expectation of how one will be treated if one’s status is 48 disclosed or becomes public), internalised or self-stigma (the adoption of a negative self-49 image and devaluation that results from living with HIV), and perceived stigma, also 50 known as community stigma (perceptions of the existence and severity of stigmatising 51 attitudes in the community that lead to anticipated stigma) [6, 7, 8]. Evidence suggests 52 that perceived stigma, or the fear of being stigmatised is the most prevalent form of stigma 53 in the era of antiretroviral expansion [9].”

·        I am afraid the authors do not get this correctly. Enacted stigma is similar to external stigma and refers to the experience of unfair treatment by others. It is not negative beliefs one has about oneself.

Please the theoretical framework on the concept of stigma in the following article and use it to support your explanation:

HIV Stigma and Moral Judgement: Qualitative Exploration of the Experiences of HIV Stigma and Discrimination among Married Men Living with HIV in Yogyakarta. International Journal of Environmental Research and Public Health. 2020; 17(2):636. https://doi.org/10.3390/ijerph17020636

HIV stigma is aggravated by variation of the effects of stigma across cultural and 55 social groups [5, 7, 10], hence the pockets of HIV related stigma observed in many settings. Thus, stigma is experienced in different ways and at different rates even within geographic locations [7, 11, 12].”

·        Here you are talking about HIV stigma in different settings:

Please consult the following resource:

Stigma and Discrimination towards People Living with HIV in the Context of Families, Communities, and Healthcare Settings: A Qualitative Study in Indonesia. International Journal of Environmental Research and Public Health. 2021; 18(10):5424. https://doi.org/10.3390/ijerph18105424

The consequence of persistent high levels of anticipated stigma within communities 63 is PLHIV hiding their HIV status from others [7, 9]. All forms of stigma have been repeatedly linked with decreased voluntary HIV testing, non-disclosure of HIV status, and un-65 willingness to disclose [17, 18, 19, 20]. Several research studies identified internalised stigma as a key risk factor for more severe negative outcomes than other forms of stigma amongst APHIV, including non-adherence to antiretroviral therapy (ART), disengagement with care, non-disclosure, and morbidity [12, 21, 22]. Internalised stigma was also found to be a stronger predictor of depressive symptoms than other forms [5, 15, 21, 23].”

·        What about the effect of enacted/external stigma? Please consult the following resources about the impacts of external stigma on access to HIV care:

Stigma and Discrimination towards People Living with HIV in the Context of Families, Communities, and Healthcare Settings: A Qualitative Study in Indonesia. International Journal of Environmental Research and Public Health. 2021; 18(10):5424. https://doi.org/10.3390/ijerph18105424

HIV Stigma and Discrimination: Perspectives and Personal Experiences of Healthcare Providers in Yogyakarta and Belu, Indonesia. Front. Med. 8:625787. doi: 10.3389/fmed.2021.625787

“On the contrary, research data about the HIV stigma perceptions, attitudes, and discrimination among the general population are limited [38]”.

·        Please revise this sentence, there have been a huge number of studies about HIV stigma perceptions, attitudes, and discrimination among the general population around the world.

The knowledge for the first objective is not convincing. Please revisit.

Discussion:

This section is very descriptive. The authors consulted the previous findings and compare their findings to those of previous studies but not much discussion is provided. I would suggest the authors revise the discussion section and try to move from description to explanatory level. For example, if you stated your findings support the previous findings, then what would be the possible explanation/reasons for that similarity? Or if you state the same discriminatory and stigmatizing attitudes and behaviours toward PLHIV also exist in other countries, then why is that? What would be the reasons or explanation for it?

Need to pay attention to the clarity of the sentences.

Author Response

We thank the reviewer for their valuable comments, all revisions are outlined below and highlighted in the document

Reviewer 3 Report

Thank you for a well written article focusing on an interesting and very important topic.  Over all the article is very well written and provides information on stigma, which is important for the design of any intervention related to HIV. 

My main question/concern is related to your measures:

1) It is unclear which reply options people had. This needs to be described more clearly. These four are mentioned in the text:   

- agree

- not sure

- disagree

- unsure

2) The grouping of replies:

L170-174 - you explain that you have chosen to code the answers dichotomously  by merge the 'unsure/not sure' with the 'agree' replies. Does this mean that an 'not sure' reply has as much weight as the 'agree' in the analysis?  If so I think this will schrew the analysis toward a conclusion of more stigmatization than actually warranted. What if you have chosen to add all 'unsure' to the 'disagree' replies?   Then the data would show less stigmatizing actitudes?

I feel the validation of this approach must be explained, clarified and justified in the text. Maybe also with some language on the potential problems with this approach. 

Minor language edits: 

L132: previously used twice

L191-195: the use of the expression: 'people who are HIV' may be a valid spoken language term, but sounds a bit discriminatory in writing. 

L221-222: 'not sure' listed twice

L237: may be ad a 'had' - , and 87% had ever tested....'

Tables: 

- would it make sense to list the statements in order with the ones most agree with on top etc?

very fine - only few edits

Author Response

We appreciate the insightful comments from the reviewer

Round 2

Reviewer 1 Report

My comments were corrected

Reviewer 2 Report

The authors have sufficiently addressed the concerns I raised in the previous round. 

There is a need for language editing.